# Validity and reliability of portable A-mode ultrasound in measuring body fat percentage: A systematic review with meta-analysis

Luiz Fernando Ferreira[1], Elirez Bezerra da Silva[2], Alexander Barreiros Cardoso Bomfim[1]*

**1** Postgraduate Program in Operational Human Performance, Brazilian Air Force University, Rio de Janeiro, Brazil, **2** Postgraduate Program in Exercise and Sport Sciences, State University of Rio de Janeiro, Rio de Janeiro, Brazil

* alexanderabcb@gmail.com

**Data Availability Statement:** All relevant data are within the paper and its Supporting Information files.

## Abstract

The present Systematic Review with Meta-analysis study aimed to evaluate the validity and reliability of the Portable A-mode Ultrasound (P-US) for measuring body fat percentage (BF %). Only studies with participants of both genders which had assessed BF% using P-US compared to the reference standard were selected. Publications up until May 31, 2022 were searched in the MEDLINE, COCHRANE, Science Direct, Web of Science, LILACS, SciELO, PEDro, SPORT Discus, CINAHL and SCOPUS databases. QUADAS-2 was used to assess the risk of bias in the validity studies and QAREL was used for the methodological quality of reliability studies. The JAMOVI software program synthesized the results, from which the Pearson Correlation Coefficient (r) or the square root of the Multiple Linear Regression Determination Coefficient ($R^2$) were extracted for the validity studies, and the Mean of Errors of the Bland-Altman Test (ME) and the Confidence Interval (95%CI) with Upper and Lower Limits for the reliability studies. A total of 13 studies were included, generating 26 results for the quantitative synthesis, 14 for validity and 12 for reliability. Regarding the validity results, a strong correlation was identified between the equipment (r = 0.870 [0.845–0.895], P<0.001), with moderate and true heterogeneity ($I^2$ = 53.47%, P = 0.003), presenting publication bias. A small effect size was identified regarding the reliability results, overestimating the results due to chance between the devices (ME = 0.207 [-0.798–1.212], P = 0.686), with low heterogeneity also due to chance ($I^2$ = 19.44%, P = 0.253), with no publication bias. All of the evaluated studies showed some violation of the instruments, confirming the high risk of bias and the low methodological quality. There is concern with heterogeneity for the validity results explained by the subgroups' analysis. The P-US associated with anthropometric perimeters satisfactorily measures the BF% with samples greater than 100 participants, and males. The results in the reliability assessment show high agreement and high variability, greatly expanding the confidence interval, which should be viewed with reservations. This review received financial support from the Brazilian Air Force. The study was registered with PROSPERO under the number CRD42020166617.

**Funding:** AB and LF received financial support from the Brazilian Air Force. The funders had no role in study design, data collection and analysis, decision to publish, or preparation of the manuscript.

**Competing interests:** The authors have declared that no competing interests exist.

## Introduction

Studies prove that physical fitness is extremely important in the success of military operations. Physically active soldiers can fight regardless of the time of day and can withstand stress in the most varied conditions, in addition to recovering more quickly from injuries when compared to those who are not physically fit [1].

Overweight/obesity was the occurrence with the greatest effect in the US Air Force (USAF) and one of the predictors associated with low physical fitness among men [2], while studies in the US Army (US-ARMY) report that obese individuals are not fit for military service [3].

Accurately determining body composition is of extreme interest to the Armed Forces. Excess adipose tissue and a high fat percentage are related to increased risk of morbidity and mortality, and the same is true among military personnel [4].

Assessment methods such as Dual-energy X-ray Absorptiometry (DXA), Air Displacement Plethysmography (ADP) and Hydrostatic Weighing (HW) can provide accurate results, but such methods are expensive and are only available in controlled environments, which makes access and operability difficult [5]. The challenge is to adopt methods with easy measurement, low cost, and which are valid and reliable, in addition to logistic feasibility for large populations and in different scenarios.

The ultrasonic evaluation method has been used for decades, showing high reliability for measuring subcutaneous fat [6] and is indicated as a practical and low-cost tool for analyzing body composition [7, 8]. Portable A-mode Ultrasound (P-US) from the Bodymetrix® brand (IntelaMetrix, Inc.–model: BX 2000) has shown promising results in terms of validity and reliability, depending on the group selected to perform the validation procedure [9].

The validity and reliability of the P-US were confirmed in studies by Johnson et al. [10], Bielemann et al. [11], Wagner; Cain; Clark [12], Schoenfeld et al. [8], Ripka et al. [7, 13] and Totosy de Zepetnek et al. [14]. Validity was only confirmed in the studies by Johnson et al. [15] and Baranauskas et al. [16], and reliability was only confirmed in the studies by Smith-Ryan et al. [9] and Hendrickson et al. [17].

The present Systematic Review with Meta-analysis is justified by the possibility of adopting an instrument for measuring body composition which is easy to use, low cost, valid, reliable, and capable of carrying out measurements in uncontrolled environments and large populations, such as in a military group.

It also aims to establish a true and temporary result from studies that presented conflicting results, such as Smith-Ryan et al. [9] or Loenneke et al. [18], which did not confirm the validity for the sample of people with overweight/obesity and with normal weight, respectively, in addition to not existing in previous reviews determining the validity and reliability of the P-US to measure body fat percentage (BF%).

## Objective

The objective of this study is to evaluate the validity and reliability of the Portable A-mode Ultrasound (P-US) for measuring body fat percentage (BF%).

## Method

This Systematic Review with Meta-analysis was written according to the PRISMA 2020 Statement [19, 20], (S1 Checklist), which suggests essential items that should be described in systematic reviews with or without meta-analysis.

## Eligibility criteria

Studies were selected considering the PIRO [21] strategy with participants of both genders, of any age (Participant), who had their body composition assessed by the P-US of the Bodymetrix® brand (Index Test) in comparison with the DXA, or the ADP or even the HW (Reference Standard) and which presented results for the BF% (Outcome).

Equipment such as DXA, ADP and HW are established as valid and reliable instruments for assessing body composition, and in particular BF%. We recognize that there is still no consensus in the literature regarding the reliability between the methods, in particular, DXA and ADP [22–25]. In the present review, we considered the results of the outcome valid and reliable among themselves.

## Sources of information and search strategy

The following databases were consulted: MEDLINE (US National Library of Medicine), COCHRANE, Science Direct, Web of Science, LILACS (Latin American and Caribbean Literature in Health Sciences), SciELO, PEDro (Physiotherapy Evidence Database), SPORT Discus, CINAHL (Cumulative Index to Nursing and Allied Health Literature) and SCOPUS from June 13 to August 13, 2019, and updated from May 22 to 31, 2022, by two researchers (LF, AB) independently. They defined the best search phrase in each database by consensus as the one with the highest number of records based on the eligibility criteria, without time or language filter. The results are shown in Fig 1.

The following search terms and their synonyms were adopted, present in the Descriptors in Health Sciences (DeCS) and the Medical Subject Headings (MeSH): "Body Composition" OR "Composition, Body" AND "US portable" OR "Ultrasound portable" OR Bodymetrix. The search phrase was obtained using the Boolean operators "AND" between the descriptors and "OR" between the synonyms. Reference lists and other sources were explored (S1 File).

## The study selection process, data collection and collected items

Two reviewers independently (LF, AB) selected the studies and disagreements were resolved by consensus. All studies found in the identification stage went through the selection phase when duplicates were manually removed using the EndNote Web software program. After reading the titles and abstracts, those that did not meet the eligibility criteria were discarded. In the impossibility of deciding on eligibility for titles and abstracts, readings of the full text were performed for selecting studies included in this review.

Data extraction was performed by two researchers (LF, AB) independently, and discrepancies were resolved by consensus. The information collected was as follows: Author(s); Kind of study; Sample characteristic; Reference standard; Protocol used / Sites measured by P-US; Result (BF%); Statistical test; Outcome result: Pearson's Correlation Test (r) or Multiple/Simple Linear Regression Determination Coefficient ($R^2$) and Bland-Altman Test values.

When results were not available by gender (M and F), the total result (M/F) was extracted. No restrictions were placed regarding when the authors declared the number of sites evaluated, the use of the P-US own analysis software or the Multiple/Simple Linear Regression Equation, but they had to be validity and/or reliability studies.

The study by Smith-Ryan et al. [9], Baranauskas et al. [16] and the study by Bielemann et al. [11] received funding from funding agencies in their countries. In addition to the available information listed above, data were collected as part of the sample's professional activity, when available.

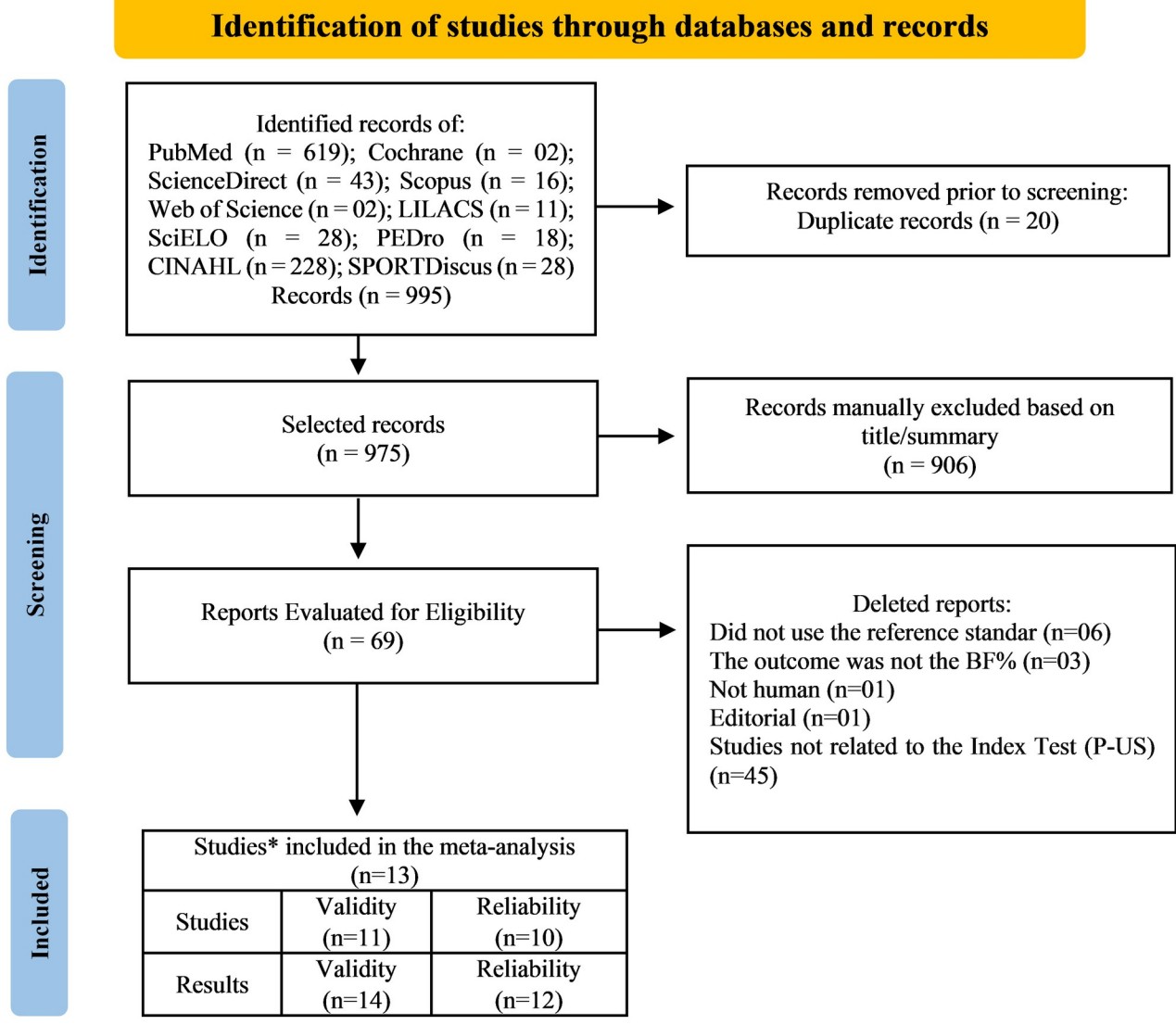

**Fig 1. Flowchart of the 13 studies included in the meta-analysis with a total of 26 results, with 14 results for validity and 12 results for reliability.**
* Studies present at least one result for validity or reliability.

### Assessment of methodological quality and risk of bias of studies

QUADAS-2 [26] was used to assess the risk of bias in the validity studies, which is structured so that the key domains (Patient Selection; Index Test(s); Reference Standard; Flow and Time) are classified in terms of risk of bias and research applicability concerns. Each key domain presents a series of guiding questions that help in the judgment of biases and applicability, as shown in Fig 2.

First, two researchers (LF, AB) performed an independent judgment through the question "What is the validity of portable ultrasound in measuring body fat percentage?" after the process of adapting QUADAS-2 to the Systematic Review. Then they recorded the information used to conclude the trial and disagreements were resolved by consensus.

In assessing the risk of bias for each of the four key domains, the guiding questions were answered as "YES", "NO" or "UNCERTAIN". All "YES" answers mean low risk of bias, while

| Study | Risk of Bias | | | | Applicability Concerns | | |
|---|---|---|---|---|---|---|---|
| | Patient Selection | Index Test | Reference Standard | Flow and Timing | Patient Selecion | Index test | Referenc Standard |
| BARANAUSKAS et al., (2015) | 2 | 1 | 1 | 3 | 2 | 1 | 1 |
| BIELEMANN et al., (2016) | 2 | 2 | 1 | 3 | 1 | 1 | 1 |
| JOHNSON et al., (2012) | 2 | 2 | 1 | 1 | 2 | 1 | 1 |
| JOHNSON et al., (2014) | 2 | 2 | 1 | 3 | 2 | 1 | 1 |
| KANG et al., (2020) | 2 | 2 | 1 | 3 | 1 | 1 | 1 |
| LOENNEKE et al., (2014b) | 2 | 1 | 1 | 3 | 2 | 1 | 1 |
| RIPKA et al., (2016a) | 2 | 1 | 1 | 3 | 2 | 1 | 1 |
| RIPKA et al., (2016b) | 2 | 2 | 1 | 3 | 2 | 1 | 1 |
| SCHOENFELD et al., (2017) | 2 | 2 | 1 | 1 | 2 | 1 | 1 |
| TOTOSY DE ZEPETNEK et al., (2021) | 2 | 2 | 1 | 1 | 1 | 1 | 1 |
| WAGNER; CAIN; CLARK, (2016) | 2 | 2 | 1 | 1 | 1 | 1 | 1 |

| Legend: | 1 | LOW RISK | 2 | HIGH RISK | 3 | UNCLEAR RISK |
|---|---|---|---|---|---|---|

**Fig 2. Methodological quality and risk of bias for validity studies according to the QUADAS-2 scale.**

if any question is answered "NO", it changes the risk of bias. At the end of the responses, the risk of bias was judged as "LOW", "HIGH" or "UNCERTAIN", without assigning scores. Applicability was also judged as "LOW", "HIGH" or "UNCERTAIN".

The QAREL tool [27] was used to independently assess the methodological quality of reliability studies by the researchers (LF, AB) and differences were resolved by consensus. There was an adaptation of the QAREL because only reproducibility studies between equipment were evaluated. Thus, questions 3 to 8 were not applied to the study and were suppressed because they are restricted to inter-rater analysis, which is not the subject of this review.

The QAREL tool consists of an 11-item checklist that considers sampling bias and representativeness of subjects and evaluators, evaluator blinding, examination order effects (order in which evaluators or subjects were examined), adequacy of the time between repeated measurements, application and appropriate test interpretation, and whether statistical analysis of reliability was appropriate, as shown in Table 1.

Each item can be answered as "YES", "NO" or "UNCERTAIN" in its evaluation, and some items include the option "NOT APPLICABLE". The questions were formulated in such a way that a "YES" answer indicates good quality for the study, while a "NO" answer indicates low quality. Those items classified as "NO" may invalidate and compromise the study results, regardless of the number of items that were classified as "YES".

### Effect measures

The results of Pearson's Correlation Coefficient (r) or the Multiple/Simple Linear Regression Determination Coefficient ($R^2$) were extracted for validity studies, and the square root was extracted, interpreted according to Mukaka [28].

**Table 1. Methodological quality and risk of bias for reliability studies according to the QAREL scale.**

| Author(s) (year) / Item | 1 | 2 | 3 | 4 | 5 | 6 | 7 | 8 | 9 | 10 | 11 |
|---|---|---|---|---|---|---|---|---|---|---|---|
| BIELEMANN et al., (2016) | N | U | X | X | X | X | X | X | U | Y | Y |
| HENDRICKSON et al., (2019) | N | Y | X | X | X | X | X | X | Y | N | Y |
| JOHNSON et al., (2014) | N | U | X | X | X | X | X | X | Y | Y | Y |
| KANG et al., (2020) [45] | N | U | X | X | X | X | X | X | U | Y | Y |
| RIPKA et al., (2016a) | N | U | X | X | X | X | X | X | U | N | Y |
| RIPKA et al., (2016b) | N | U | X | X | X | X | X | X | U | Y | Y |
| SCHOENFELD et al., (2017) | N | Y | X | X | X | X | X | X | U | N | Y |
| SMITH-RYAN et al., (2014) | N | Y | X | X | X | X | X | X | Y | N | Y |
| TOTOSY DE ZEPETNEK et al., (2021) | N | U | X | X | X | X | X | X | Y | N | Y |
| WAGNER; CAIN; CLARK, (2016) | N | Y | X | X | X | X | X | X | Y | N | Y |

Y–Yes; N–No; U–Uncertain; N/A - Not Applicable; X–Questions excluded; 1 –Sampling Bias; 2 –Experience of the Evaluators; 3 to 7 –Blinding of the Evaluators; 8 – Reliability between Evaluators; 9 –Interval and Time; 10 –Test Application and Interpretation; 11 –Statistical Test.

Next, the results of the Bland-Altman Test [29], mean of errors (ME), the standard deviation of errors (SDe), upper (UL) and lower (LL) limits with a 95% confidence interval (95%CI) were extracted from the reliability studies to assess the agreement between the measurements measured by the Standard Reference equipment and the P-US; however, when no information was provided in the text, data were imputed from the graph through the Meazure version 2.0.1 software program.

The Bland-Altman Test is widely used in the medical sciences to verify the agreement between two measurements, a new measurement and the reference standard measurement, which is generally more expensive or invasive [30].

The analysis performed by the Bland-Altman presents a graphical variation of dispersion which enables evaluating the agreement between the methods, with the mean value between the measures being on the X axis [(P-US + REFERENCE STANDARD)/2], and the error obtained from the difference between the fat percentage measurements being on the Y axis (P-US–REFERENCE STANDARD), resulting in the ME and the 95%CI (ME ± 1.96SDe), as indicated in the graph [31].

Reliability results in repeatability (intra-evaluator) and reproducibility (inter-evaluator) were discarded. There is consensus in the literature that the P-US is capable of producing reliable results when the evaluators are experienced and have undergone training, as according to the studies by Smith-Ryan et al. [9], Wagner; Cain, Clark [12], Hendrickson et al. [17], Totosy de Zepetnek et al. [14], Ferreira and Bomfim [32], and Carvalho et al. [33].

## Synthesis of collected data methods

A summary table was constructed (Table 2) with the identification of the studies, the type of study, the sample characteristics, as well as the outcomes for validity and reliability. Data analysis was performed by an independent researcher (LF) using the JAMOVI software program.

The analysis in validity studies was performed using Pearson's Correlation Coefficient (r), crude correlation as a continuous outcome measure. The Q Test [34] and the $I^2$ statistic were used to assess heterogeneity ($tau^2$), and the Z Test to estimate the 95% confidence interval. A statistical significance level of $P \leq 0.05$ was adopted (Fig 3).

The analysis in reliability studies was performed using the effect size from the ME and SDe values of the Bland-Altman Test as a continuous outcome measure. The Q Test [34] and the $I^2$

**Table 2. Data extracted from the studies included in the meta-analysis.**

| Author(s) (year) | Study type | Sample characteristics | Reference standard | Protocol used / Sites measured | Result (BF%) | Statistical test | Outcome | |
|---|---|---|---|---|---|---|---|---|
| | | | | | | | r | Bland-Altman ME±SDe (UL–LL) |
| BARANAUSKAS et al., (2015) | Validity | n = 76 (33M and 43F) Age: 22.8 ± 2.5 years Mass: 70.35 ± 16.12 kg Height: 1.71 ± 0.12 m | DXA | BodyView / JP7 | DXA 25.19 ± 8.62 P-US 21.31 ± 6.29 | Pearson's Correlation | 0.816 | # |
| BIELEMANN et al., (2016) | Validity and Reliability | n = 104 (F) Age: 31.9 ± 9.9 years BMI: 24.8 ± 4.3 Kg/m² | ADP | **Sites (SK):** Abdominal, Calf **Circunference:** Calf | ADP 34.1 ± 8.0 P-US 34.2 ± 7.3 | Multiple Linear Regression Bland-Altman | 0.901* | 0.1±3.37 (-6.5–6.7) |
| | | n = 102 (M) Age: 30.0 ± 8.1 years BMI: 25.7 ± 3.7 Kg/m² | ADP | **Sites (SK):** Triceps, Thigh **Circunference:** Thigh **Diameter:** Biceps muscle | ADP 21.2 ± 7.3 P-US 21.7 ± 6.4 | Multiple Linear Regression Bland-Altman | 0.854* | 0.5±3.67 (-6.8–7.7) |
| HENDRICKSON et al., (2019) | Reliability | n = 31 (21M and 10F) Age: 26.7.8 ± 3.9 years Mass: 75.0 ± 13.8 kg Height: 1.77 ± 0.9 m BMI: 23.9 ± 3.0 Kg/m² | ADP | BodyView / JP3 | ADP # P-US 17.6 ± 6.9 | Bland-Altman | # | 0±3.57 (-9–7) |
| JOHNSON et al., (2012) | Validity | n = 26 (18M and 8F) Age: 22.9 ± 1.35 years | ADP | BodyView / JP3 | ADP 15.5 ± 5.83 P-US 15.7 ± 5.14 | Pearson's Correlation | 0.879 | # |
| JOHNSON et al., (2014) | Reliability | n = 84 (35M and 49F) Age: 23.0 ± 4.7 years Mass: 70.35 ± 15.97 kg Height: 1.71 ± 0.12 m BMI: 23.6 ± 3.6 Kg/m² | DXA | BodyView / JP7 | DXA 25.6 ± 0.9 P-US 21.6 ± 0.76 | Bland-Altman | # | 4.4±4.8 (-5.1–13.8) |
| | Validity | n = 49 (F) Age: 23.0 ± 4.7 years Mass: 60.55 ± 10.30 kg Height: 1.64 ± 0.09 m BMI: 22.0 ± 3.1 Kg/m² | DXA | BodyView / JP7 | DXA 29.7 ± 1.0 P-US 26.0 ± 0.5 | Pearson's Correlation | 0.65 | # |
| | | n = 35 (M) Age: 23.0 ± 3.3 years Mass: 83.73 ± 11.97 kg Height: 1.80 ± 0.08 m BMI: 23.6 ± 3.2 Kg/m² | DXA | BodyView / JP7 | DXA 20.1 ± 1.3 P-US 15.7 ± 0.6 | Pearson's Correlation | 0.84 | # |
| KANG et al., (2020) | Validity and Reliability | n = 105 (M) Age: 20.01 ± 2.11 years Mass: 73.26 ± 13.6 kg Height: 1.72 ± 0.10 m BMI: 23.91 ± 3.77 Kg/m² | DXA | BodyView / 9 Parllo | DXA 18.22 ± 8.4 P-US 17.77 ± 3.80 | Pearson's Correlation Bland-Altman | 0.790 | 0.4 ± 5.56 (-10.4–11.3) |
| LOENNEKE et al., (2014) | Validity | n = 13 (F) Age: 20 ± 1 years Mass: 62.9 ± 7.6 kg Height: 1.60 ± 0.1 m | DXA | BodyView / JP3 | DXA 21.6 ± 5.7 P-US 25.0 ± 4.4 | Pearson's Correlation | 0.753 | # |
| RIPKA et al., (2016a) | Validity and Reliability | n = 143 (M) Age: 14.77 ± 1.49 years Mass: 60.45 ± 11.05 kg Height: 1.69 ± 0.09 m | DXA | **Sites (SK):** Triceps, Thigh, Suprailiac | DXA 19.94 ± 5.36 P-US 10.57 ± 4.45 | Multiple Linear Regression Bland-Altman | 0.905* | 0.15±1.19 (-4.72–2.49) |

*(Continued)*

**Table 2.** (Continued)

| Author(s) (year) | Study type | Sample characteristics | Reference standard | Protocol used / Sites measured | Result (BF%) | Statistical test | Outcome | |
|---|---|---|---|---|---|---|---|---|
| | | | | | | | r | Bland-Altman ME±SDe (UL–LL) |
| RIPKA et al., (2016b) | Validity and Reliability | n = 34 (F) Age: 13.0 ± 2.3 years Mass: 52.8 ± 9.3 kg BMI: 20.82 ± 4.31 Kg/m² | DXA | **Sites (SK):** Triceps, Subscapularis, Thigh, Chest | DXA 30.3 ± 4.9 P-US 22.5 ± 5.7 | Multiple Linear Regression Bland-Altman | 0.844* | 0±3.57 (-7.0–7.0) |
| | | n = 71 (M) Age: 14.0 ± 2.0 years Mass: 57.8 ± 12.2 kg BMI: 20.79 ± 3.11 Kg/m² | DXA | **Sites (SK):** Triceps, Subscapularis, Thigh, Chest | DXA 20.0 ± 7.2 P-US 9.6 ± 6.6 | Multiple Linear Regression Bland-Altman | 0.921* | 0±2.96 (-5.4–5.8) |
| SCHOENFELD et al., (2017) | Validity and Reliability | n = 20 (F) Age: 22.4 ± 2.8 years Mass: 62.2 ± 6.5 kg Height: 1.63 ± 0.05 m | ADP | BodyView / JP4 | ADP 25.59 ± 7.94 P-US 26.51 ± 6.32 | Pearson's Correlation Bland-Altman | 0.86 | -0.92±4.09 (-9.0–7.09) |
| SMITH-RYAN et al., (2014) | Reliability | n = 47 (20M and 27F) Age: 37.6 ± 11.6 years Mass: 94.1 ± 16.1 kg Height: 1.73 ± 0.10 m BMI: 31.5 ± 5.2 Kg/m² | ADP | BodyView / JP7 | ADP 33.7 ± 7.6 P-US 29.0 ± 6.5 | Bland-Altman | # | -4.72**±3.73 (-12.0–2.59) |
| TOTOSY DE ZEPETNEK et al., (2021) | Validity and Reliability | n = 49 (16M and 33F) Age: 31.4 ± 10.7 years Mass: 68.0 ± 12.9 kg Height: 1.69 ± 0.8 m BMI: 23.5 ± 3.0 Kg/m² | ADP | **Sites (SK):** (M/F) Chest, Abdominal, Thigh, Triceps, Subscapularis, Suprailiac, Mid Axillary | ADP 24.7 ± 7.2 P-US 24.3 ± 6.6 | Multiple Linear Regression Bland-Altman | 0.848* | - 0.32±3.85 (-7.87–7.22) |
| WAGNER; CAIN; CLARK, (2016) | Validity and Reliability | n = 45 (22M and 23F) Age: 20.1 ± 1.6 years Mass: 71.8 ± 12.4 kg Height: 1.72 ± 0.10 m BMI: 24.1 ± 2.4 Kg/m² | ADP | **Sites (SK):** (M) Chest, Abdominal, Thigh (F) Triceps, Thigh, Suprailiac | ADP 14.9 ± 6.7 P-US 18.2 ± 7.6 | Simple Linear Regression Bland-Altman | 0.921* | 3.2**±2.96 (-2.8–9.0) |

n–Number of evaluated subjects; (M)–Male; (F)–Female; BF%–Body Fat Percentage; P-US–Portable Ultrasound; BodyView–P-US analysis software; DXA–Dual-energy X-ray Absorptiometry; JP7 –Jackson–Pollock Protocol, Seven Sites; JP4 - Jackson–Pollock Protocol, Four Sites; JP3 - Jackson–Pollock Protocol, Three Sites; L3 –Lohman Protocol, Three Sites; 9 Parllo–Parllo Protocol, 9 sites; r–Pearson's Correlation Coefficient; LL–Lower Limit; UL–Upper Limit; ME–Mean of Errors of the Bland-Altman Test; SDe–Standard Deviation of Errors of Bland-Altman Test [(Bland-Altman UL-ME)/1.96]; BMI–Body Mass Index; ADP–Air Displacement Plethysmography; Sites (SK): skinfolds measured by P-US; (*) extracted the square root of the results of the Multiple/Simple Linear Regression Determination Coefficient; (**) Imputed data.

statistic were also used to assess heterogeneity (tau²), and the Z Test to estimate the confidence interval (CI) at 95%. A statistical significance level of P≤0.05 was adopted (Fig 5).

A funnel plot and the Egger test were used to assess publication bias, adopting a statistical significance level of P≤0.05. P-values of the Egger's Test lower than 0.05 and the absence of the isosceles triangle in the funnel plot characterize publication bias (Figs 4 and 6).

Men and women have different body composition characteristics [35], affecting the results of the outcomes, especially for those methods that use subcutaneous adipose tissue to measure BF%, such as P-US.

Validity studies with P-US can be performed by associating the BF% value measured through its analysis software program (BodyView) with the reference standard, characterized as simple regression studies. Other studies can be performed through the value expressed in millimeters measured by P-US at different anthropometric sites, combined with perimeters,

diameters, among other predictive body variables, to predict the BF% result of the reference standard, characterized as multiple regression studies.

Thus, the analysis of subgroups was adopted in the following extracts to detect the cause of heterogeneity: Male, Female, Total (Male and Female), Simple Regression and Multiple Regression, as shown in Table 3.

Studies that did not present adequate statistical tests for the selected outcomes were excluded. The retrieved studies invariably presented more than one result for both validity and reliability. The extracted and selected results obeyed the following criteria in this order: the Multiple Linear Regression Determination Coefficient with the highest number of sites evaluated by P-US, the highest value of associations (r) and the lowest ME value of the Bland-Altman Test.

## Results

The studies by Ulbricht et al. [36], Neves et al. [37], Loenneke et al. [38], Chiriţă-Emandi et al. [39], Krkeljas et al. [40] and Elsey et al. [41] were excluded because they did not compare the P-US with the DXA, the ADP or the WH; Utter and Hager [42], Johnson, et al. [43] and Jones [44] were excluded because the outcome was not BF%, according to the PIRO [21] strategy proposed in this Systematic Review.

### Validity studies

See Figs 3 and 4.

### Reliability studies

See Figs 5 and 6.

## Discussion

The main findings of this Systematic Review with Meta-analysis are: (1) the Validity of the P-US was confirmed through a strong correlation with the reference standard; and (2) the Reliability was confirmed due to the low bias found between the values achieved by the P-US and the reference standard.

### Validity studies

All studies showed at least one violation of the instrument regarding the key domain "Patient Selection", characterizing it as at high risk of bias (Fig 2). Violations of the instrument were restricted to the sample, such as the lack of criteria declared by the authors for selection, size and composition and the lack of reporting on the interval time between one test and another.

The outcome result (BF%) for validity presented by the 11 studies included with 14 results meta-analyzed by the JAMOVI software program (Version 2.2.5) was reported by Pearson's correlation coefficient (Fig 3). The result of the meta-analysis was 0.870 with a 95% CI between [0.845–0.895], with P <0.001, which indicates a strong correlation between the P-US and the reference standard [28].

The present meta-analysis also shows moderate heterogeneity of 53.47% and true (P = 0.003), which justifies the random effect analysis model with the Hunter-Schmidt protocol [46, 47], with heterogeneity being incorporated in the result of the meta-analysis.

The analysis to explain the heterogeneity was performed by subgroups: male/female, male only, female only, simple regression studies and multiple regression studies (Table 3). The

results by subgroups point to a low to moderate heterogeneity, indicating the fixed effect analysis model when P>0.05.

From the data presented (see Table 3), the Male Subgroup (Subgroup [M]) has similar inconsistency to the other groups, despite having twice the sample size than the other extracts (Subgroup [M/F] and Subgroup [F]), which can be explained by the difficulty in measuring subcutaneous fat in women.

The difficulty in measuring BF% in women or obese individuals of both genders occurs in instruments that assess subcutaneous fat (skinfold caliper and P-US), crediting this difficulty to the fact that subcutaneous fat is not easily separated from the muscle [10, 48, 49].

When performing the analysis by the simple regression and multiple regression study types, the inconsistency that was at 53.47% true changed to 18.87% and 25.91%, respectively, due to probability, which confirms that the different ways of achieving the validity of the instrument influenced the results of the present meta-analysis.

The results that contributed the most and the least to the meta-analysis result relative to the weight of each study (graphically available through the forest plot) had different characteristics. Two of the four best results present samples greater than 100 participants, three of them with male components and up to three P-US measurements associated with anthropometric perimeters developed in multiple linear regression equations. Moreover, the four worst results all contain less than 50 participants, female, with three or more P-US measurements without the association with anthropometric perimeters and with the use of Pearson's Correlation Coefficient.

Despite the excellent results achieved, studies developed with anthropometric measurements associated with P-US measurements in multiple linear regression equations did not show the cross-validation results characterized by a new evaluation of the equation achieved with a smaller sample, on average 20% of the initial sample, which we consider a serious methodological error.

Due to the lack of cross-validation, the results only have internal validity and cannot be adopted for similar groups or other groups. There is concern with heterogeneity explained by the subgroups' analysis; however, the P-US measuring up to three sites, associated with anthropometric perimeters in Multiple/Simple Linear Regression Equations, with sample sizes greater than 100 participants and males seems to satisfactorily evaluate the body fat percentage.

Publication bias was confirmed in both the qualitative analysis (see Fig 4) with the absence of the funnel plot, and in the quantitative analysis with the Egger test (P<0.001). The problems reported were in the development of each study, mainly related to the size, stratification and composition of the sample, as well as the different ways of measuring the outcome, which may have influenced the publication bias.

Despite the high association found between the instruments to measure the BF%, the body fat percentage result must be evaluated together with the agreement between them, constituting the reliability analysis.

## Reliability studies

Six questions related to procedures between evaluators were excluded in assessing the methodological quality of reliability studies, as they did not fit the purpose of this meta-analysis (Table 1).

All studies showed some violation of the instrument's items, confirming the low methodological quality of the evaluated studies, which characterizes them as having a high risk of bias. The main violations were the lack of information about the evaluators' experience in

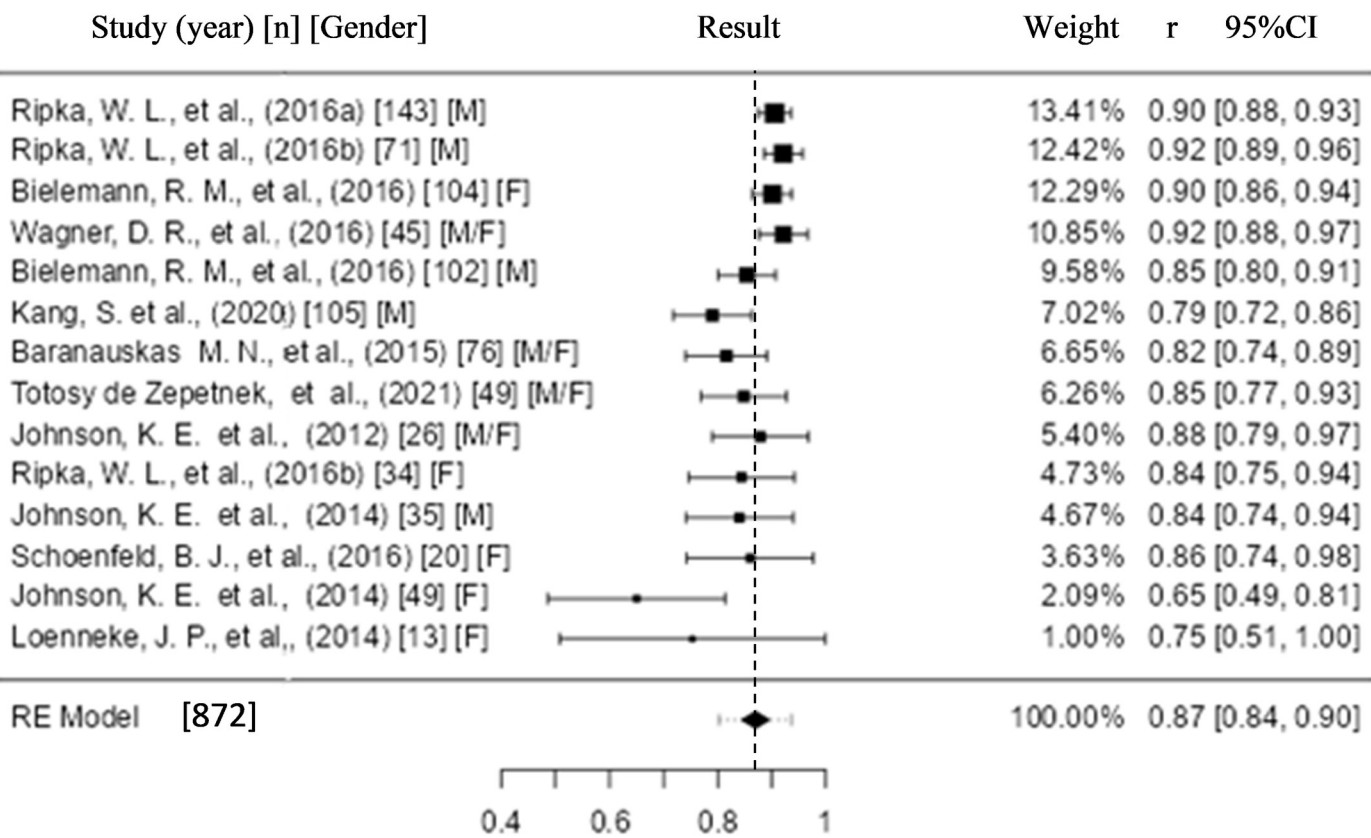

**Fig 3. Forest plot of the 11 included studies, with 14 meta-analyzed results. Result:** Random Effect–Hunter-Schmidt (k = 14): Intercept 0.870 [0.845–0.895]; P<0.001. Heterogeneity: Tau$^2$ 0.001 (SE = 7e-04); I$^2$ = 53.47%; Q = 31.502; P = 0.003.

evaluating the interval outcome between measuring the measures and the lack of adequate interpretation in the Bland-Altman Test.

The 12 results present in 10 meta-analysis studies (Fig 5) for the Bias outcome was 0.207 [-0.798–1.212], with P = 0.686, which indicates high agreement between the P-US and the reference standard, and low heterogeneity (I$^2$ = 19.44%) with P = 0.253. Even admitting the possibility of inaccuracy between the DXA and ADP instruments (admitted as the reference standard), it seems that the meta-analysis was able to mitigate the incompatibility between them.

Although the results achieved by the P-US for reliability demonstrated by the forest plot (Fig 5) are promising, they should be viewed with caution.

The individual results in each study show variations which may compromise the precision of the P-US measurement. We highlight the study by Bielemann et al. [11] (see Table 2) for males, included in this meta-analysis, which presents mean errors of 0.5 [-6.8 to 7.7]. From this result, a eutrophic man (similar to the sample of the referenced study) may present a fat percentage of 22% in the reference standard equipment, while this result by P-US may be between 15.2 and 29.7%, within the confidence interval of 95%, which we believe presents a high variation when it comes to the outcome (BF%), although the authors reported that there was no systematic bias.

Publication bias (Fig 6) was not confirmed in either the qualitative analysis (with the appearance of the isosceles triangle), nor in the quantitative analysis, with the Egger Test showing P = 0.934.

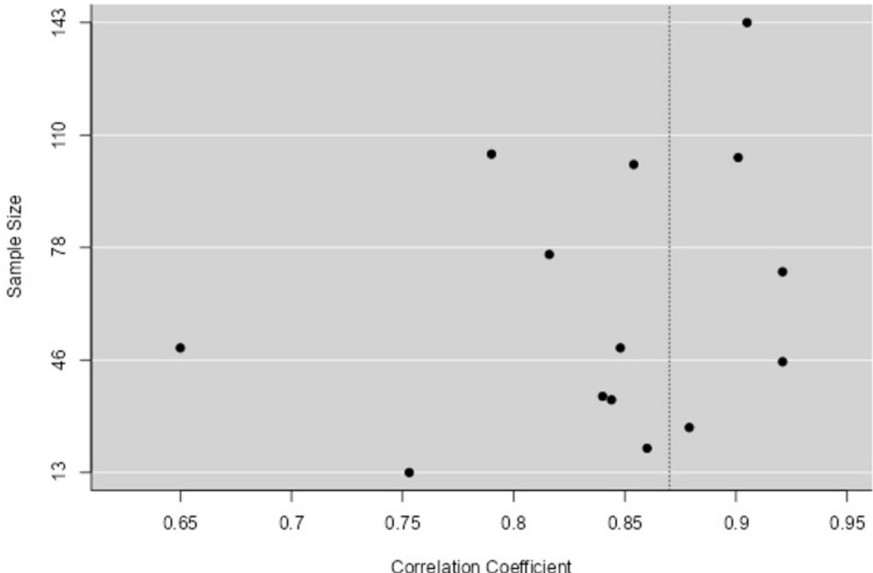

**Fig 4. Funnel plot, qualitative analysis of publication bias. Result:** The Egger Test quantitatively evaluated the risk of publication bias, and the following results were found: Egger's Test = -4.393 P <0.001.

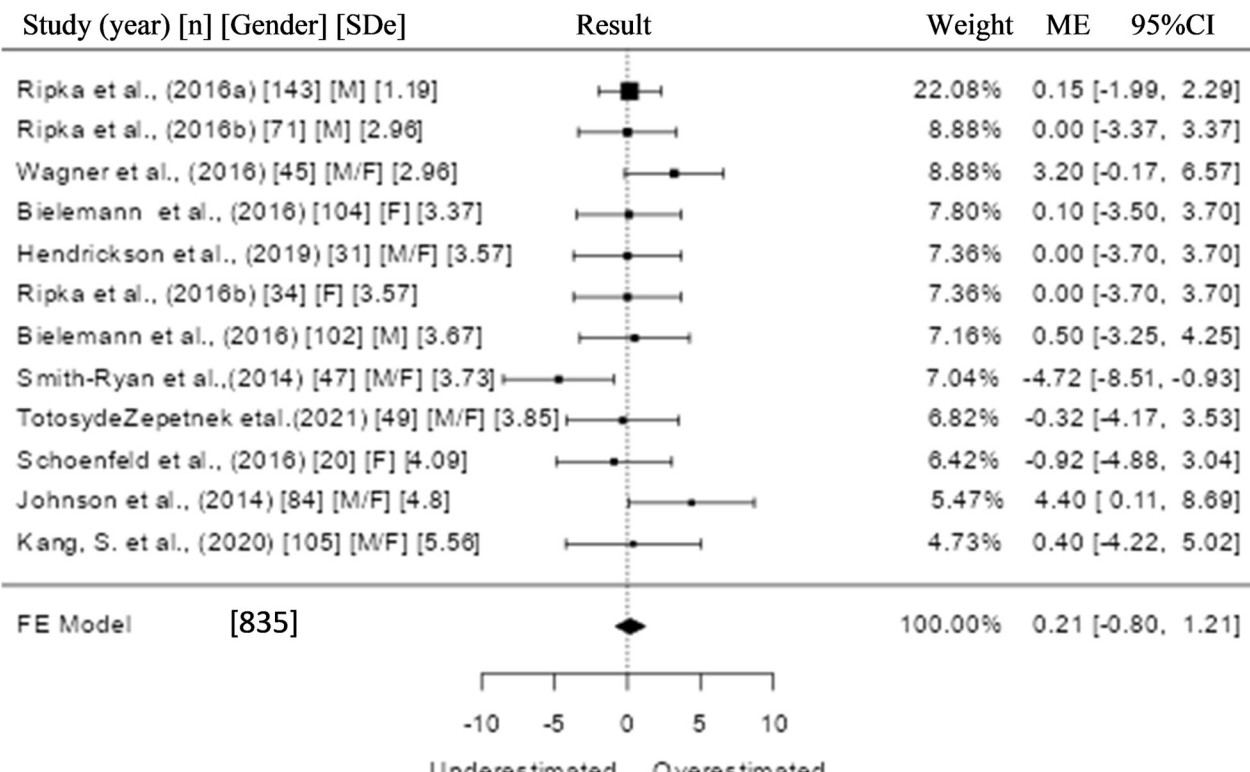

**Fig 5. Forest plot of the 10 included studies, with 12 meta-analyzed results. Result:** Fixed Effect (k = 12): Intercept 0.207 [-0.796–1.212]; P = 0.686. Heterogeneity: $Tau^2$ 0 (SE = NA); $I^2$ = 19.44%; Q = 13.655; P = 0.253.

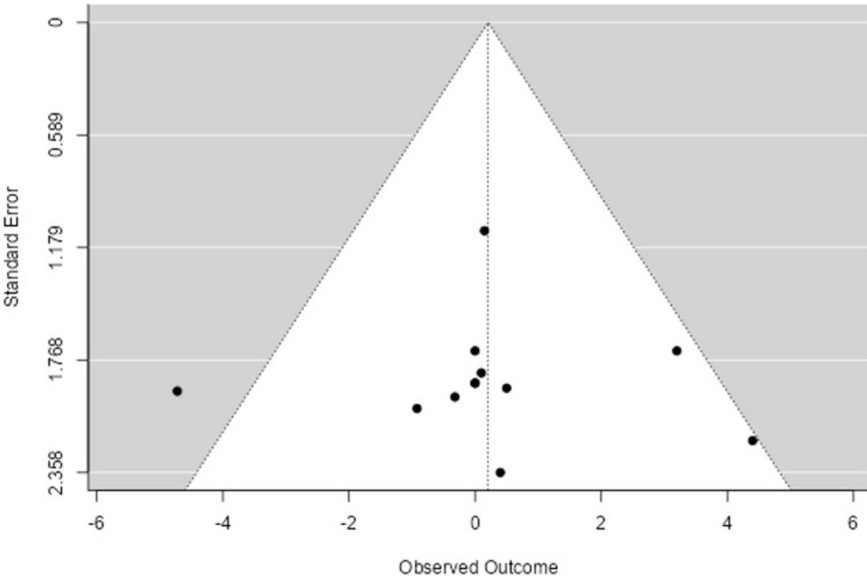

**Fig 6. Funnel plot, qualitative analysis of publication bias. Result:** The Egger test quantitatively evaluated the risk of publication bias, with the following results: Egger's Test = 0.083; P = 0.934.

The results of publication bias confirm a pattern for reliability studies involving the P-US that still falls short of reliability studies of the same nature, which can be proven by the high risk of bias and low methodological quality of the studies, as evaluated by the QAREL.

The results of the bias presented by the studies in the reliability assessment reveal high agreement, despite the variability of the results, which increases the confidence interval and the necessary clinical precision.

**Table 3. Results of subgroup analysis to detect the cause of heterogeneity.**

| Type of analysis (n) | Results included "Total" (year) [n] | Type of effect | Heterogeneity | Meta-analysis |
|---|---|---|---|---|
| **Subgroup [M/F] (196)** | "4" Baranauskas et al., (2015) [76]; Johnson et al., (2012) [26]; Totosy de Zepetnek et al., (2021) [49]; Wagner; Cain; Clark (2016) [45] | Fixed | $Tau^2 = 0$ (SE = NA) $I^2 = 54\%$ Q = 6.522; P = 0.089 | 0.884 [0.852–0.917] P < 0.001 |
| **Subgroup [F] (220)** | "5" Bielemann et al., (2016) [104]; Johnson et al., (2014) [49]; Loenneke et al., (2014b) [13]; Ripka et al., (2016b) [34]; Schoenfeld et al., (2017) [20] | Random (Hunter-Smith) | $Tau^2 = 0.0015$ (SE = 0.0012) $I^2 = 35.41\%$ Q = 10.557 P = 0.032 | 0.849 [0.792–0.906] P < 0.001 |
| **Subgroup [M] (456)** | "5" Bielemann et al., (2016) [102]; Johnson et al., (2014) [35]; Kang et al., (2020) [105]; Ripka et al., (2016a) [143]; Ripka et al., (2016b) [71] | Random (Hunter-Smith) | $Tau^2 = 9e{-}04$ (SE = 8e-04); $I^2 = 60.87\%$ Q = 14.076 P = 0.007 | 0.876 [0.840–0.912] P < 0.001 |
| **Subgroup [Simple Regression] (324)** | "9" Baranauskas, et al., (2015) [76]; Johnson et al., (2012) [26]; Johnson et al., (2014) [49]; Johnson et al., (2014) [35]; Kang et al., (2020) [105]; Loenneke et al., (2014) [13]; Schoenfeld et al., (2016) [20] | Fixed | $Tau^2 = 0$ (SE = NA); $I^2 = 18.87\%$ Q = 7.396 P = 0.286 | 0.818 [0.781–0.855] P < 0.001 |
| **Subgroup [Multiple Regression] (548)** | "5" Bielemann et al., (2016) [104]; Bielemann et al., (2016) [102]; Ripka et al., (2016a) [143]; Ripka et al., (2016b) [34]; Ripka et al., (2016b) [71]; Totosy de Zepetnek et al., (2021) [49]; Wagner et al., (2016) [45] | Fixed | $Tau^2 = 0$ (SE = NA); $I^2 = 25.91\%$ Q = 8.0098 P = 0.231 | 0.901 [0.885–0.917] P < 0.001 |

The P-US and the use of anthropometric measurements in Multiple Linear Regression Equations seem to present better results when compared with the reference standard.

The results confirm the accuracy of the equipment, similar to others which measure BF% through subcutaneous fat, such as the skinfold caliper. The ease in measuring the BF%, low cost and logistical feasibility for large populations are advantages which should be considered, proving to be effective for physical training centers, for monitoring changes in body composition, in epidemiological approaches and in evaluations that do not require clinical precision.

## Conclusion

Based on the results of this Systematic Review with Meta-analysis, the P-US is a valid and reliable device for measuring body fat percentage. Studies with representative samples of the population, selected at random in adequate amounts, preferably using a software program for this purpose or with 20 individuals for each variable evaluated are recommended, as well as studies that assess the agreement between the P-US and the reference standard method, the analysis of the results using the Bland-Altman Test and the dispersion graph between the methods, with the objective of verifying the bias behavior.

Finally, samples with different characteristics are suggested to verify the potential of P-US in measuring body fat percentage, especially samples with high deposits of subcutaneous fat. There must be a report about the interval time between one test and another, preferably less than 48 hours; in addition, the anthropometric measurements should be associated with the measurements measured by the P-US in multiple linear regression equations and with the adoption of a group for the cross-validation in order to assess the ability of the P-US to measure the body fat percentage for external validity.

## Registration and protocol

This Systematic Review with Meta-analysis was registered on the PROSPERO platform (International Prospective Register of Systematic Review), on 02/27/2020 and received the provisional ID number 166617 and the definitive registration on 04/28/2020 number CRD42020166617, updated on 07/07/2021, available at: https://www.crd.york.ac.uk/prospero/display_record.php?ID=CRD42020166617.

## Supporting information

**S1 Checklist. PRISMA 2020.**
(PDF)

**S1 File. Search phrases.**
(PDF)

**S2 File. QAREL supplement.**
(PDF)

**S3 File. QUADAS-2 supplement.**
(PDF)

**S4 File. Statistical analysis supplement.**
(ZIP)

## Author Contributions

**Conceptualization:** Luiz Fernando Ferreira, Alexander Barreiros Cardoso Bomfim.

**Data curation:** Luiz Fernando Ferreira, Alexander Barreiros Cardoso Bomfim.

**Formal analysis:** Luiz Fernando Ferreira, Elirez Bezerra da Silva, Alexander Barreiros Cardoso Bomfim.

**Funding acquisition:** Alexander Barreiros Cardoso Bomfim.

**Methodology:** Luiz Fernando Ferreira, Elirez Bezerra da Silva, Alexander Barreiros Cardoso Bomfim.

**Project administration:** Alexander Barreiros Cardoso Bomfim.

**Resources:** Luiz Fernando Ferreira, Elirez Bezerra da Silva, Alexander Barreiros Cardoso Bomfim.

**Software:** Alexander Barreiros Cardoso Bomfim.

**Supervision:** Alexander Barreiros Cardoso Bomfim.

**Visualization:** Luiz Fernando Ferreira.

**Writing – original draft:** Luiz Fernando Ferreira, Alexander Barreiros Cardoso Bomfim.

**Writing – review & editing:** Luiz Fernando Ferreira, Elirez Bezerra da Silva, Alexander Barreiros Cardoso Bomfim.

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
