## [Decision Letter · Decision Letter 0]

29 Jun 2023

PONE-D-23-06958Validity and reliability of portable ultrasound in measuring body fat percentage: A systematic review with meta-analysisPLOS ONE

Dear Dr. BOMFIM, Alexander

Thank you for submitting your manuscript to PLOS ONE. After careful consideration, we feel that it has merit but does not fully meet PLOS ONE’s publication criteria as it currently stands. Therefore, we **invite you to submit a revised version** of the manuscript that addresses the points raised during the review process.

===Academic Editor===

Dear authors,

Please read and make the review indicated or refute them with reasons.

I suggest revising the objective to:

- Evaluate the validity and reliability of the P-US for measuring body fat percentage (BF%).

Cordially

Please submit your revised manuscript **by** Aug 13 2023 11:59PM. If you will need more time than this to complete your revisions, please reply to this message or contact the journal office at plosone@plos.org. Please include the following items when submitting your revised manuscript:A rebuttal letter that responds to each point raised by the academic editor and reviewer(s). You should upload this letter as a separate file labeled 'Response to Reviewers'.A marked-up copy of your manuscript that highlights changes made to the original version. You should upload this as a separate file labeled 'Revised Manuscript with Track Changes'.An unmarked version of your revised paper without tracked changes. You should upload this as a separate file labeled 'Manuscript'.

We look forward to receiving your revised manuscript.

Kind regards,

Patricia Rezende do Prado

Academic Editor

PLOS ONE

Journal Requirements:

Reviewers' comments:

Reviewer's Responses to Questions

**Comments to the Author**

1. Is the manuscript technically sound, and do the data support the conclusions?

Reviewer #1: Yes

Reviewer #2: Yes

2. Has the statistical analysis been performed appropriately and rigorously? 

Reviewer #1: Yes

Reviewer #2: Yes

3. Have the authors made all data underlying the findings in their manuscript fully available?

Reviewer #1: No

Reviewer #2: Yes

4. Is the manuscript presented in an intelligible fashion and written in standard English?

Reviewer #1: No, review, please.

Reviewer #2: Yes

5. Review Comments to the Author

Reviewer #1: The authors made some progress in the meta-analysis and systematic review of Bodymetrix® branded portable ultrasound (P-US) measurement of body fat percentage, supplementing our previous understanding of how to use P-US to measure body fat percentage and suggesting that we have important implications for better, faster and more accurate measurement of body fat percentage.

Reviewer #2: This paper adresses an interesting problem- whether Portable Ultrasound (P-US) is a valid and reliable method to measure body fat percentage (BF%). The authors propose systematic review with Meta-analysis study to review and

synthesize the existing literatures. The results show high validity and reliability of the P-US as a device for measuring body fat percentage. Publication bias and subgroups analysis are also performed. Overall, the article is well organized and the statistical analysis is rigorous and sound.

6. PLOS authors have the option to publish the peer review history of their article (what does this mean?). If published, this will include your full peer review and any attached files.

Reviewer #1: **Yes**

Reviewer #2: No

---

## [Author Response · Author response to Decision Letter 0]

2 Aug 2023

In order to respond to the editor’s and reviewers’ suggestions, the changes and arguments regarding the comments are presented below:

Comment from the Academic Editor:

Change in the purpose of the study, “Evaluate the validity and reliability of the P-US for measuring body fat percentage (BF%)”.

Response to Academic Publisher's: In view of the comment, we decided to change the study objective, as suggested.

Reviewers' comments.

1. Is the manuscript technically sound, and do the data support the conclusions?

Reviewer #1: Yes

Reviewer #2: Yes

Response to Reviewers: no comment.

2. Has the statistical analysis been performed appropriately and rigorously?

Reviewer #1: Yes

Reviewer #2: Yes

Response to Reviewers: no comments, thanks!

3. Have the authors made all data underlying the findings in their manuscript fully available?

Reviewer #1: No

Reviewer #2: Yes

Response to Reviewers: In response to Reviewer 1's comment, we decided to insert the data sheets provided by the JAMOVI software for the validity and reliability results, the analysis of the QUADAS-2 instrument, as well as that of the QAREL (Supplement 2).

4. Is the manuscript presented in an intelligible fashion and written in standard English?

Reviewer #1: No, review, please. 

Reviewer #2: Yes

Response to Reviewers: In response to Reviewer 1's comment, we decided to forward the manuscript to a professional native English (Canadian) reviewer for further review.

5. Review Comments to the Author

Reviewer #1: The authors made some progress in the meta-analysis and systematic review of Bodymetrix® branded portable ultrasound (P-US) measurement of body fat percentage, supplementing our previous understanding of how to use P-US to measure body fat percentage and suggesting that we have important implications for better, faster and more accurate measurement of body fat percentage.

Reviewer #2: This paper addresses an interesting problem- whether Portable Ultrasound (P-US) is a valid and reliable method to measure body fat percentage (BF%). The authors propose systematic review with Meta-analysis study to review and synthesize the existing literatures. The results show high validity and reliability of the P-US as a device for measuring body fat percentage. Publication bias and subgroups analysis are also performed. Overall, the article is well organized and the statistical analysis is rigorous and sound.

Response to Reviewers: no comments, thanks!

6. PLOS authors have the option to publish the peer review history of their article (what does this mean?). If published, this will include your full peer review and any attached files

Do you want your identity to be public for this peer review? For information about this choice, including consent withdrawal, please see our Privacy Policy.

Reviewer #1: Yes

Reviewer #2: No

In response to questions raised by Reviewer 1 through the file “Peer review.docx”.

The authors made some progress in the meta-analysis and systematic review of Bodymetrix® branded portable ultrasound (P-US) measurement of body fat percentage, supplementing our previous understanding of how to use P-US to measure body fat percentage and suggesting that we have important implications for better, faster and more accurate measurement of body fat percentage. However, there are still some issues that need further improvement, as described below. 

Major Issues

1. Lines 105-108, in the data standard, what is the specific method of P-UB compared with DXA, ADP, HW methods? Are the resulting BF% values true and comparable? On what basis?

Lines 105-108: “Considering the PIRO(21) strategy, studies were selected with participants of both genders, of any age (Participant), who assessed body composition with the P-US of the Bodymetrix® brand (Index Test) in comparison with the DXA, or the ADP or even the HW (Reference Standard) and that presented results for the BF% (Outcome)”.

Reply to Reviewer 1:

In response to the question raised, we decided to characterize the way of decoding the signals emitted by the ultrasound, inserting the term “A-mode” (amplitude), characteristic of the Portable Ultrasound A-mode (US-P) of the brand Bodymetrix® (IntelaMetrix, Inc – model: BX 2000). We inserted the term “A-mode” in lines 4, 23, 75, and 95.

Although DXA, ADP, and HW are established as valid methods for assessing body composition, and in particular fat percentage, we recognize that there is still no consensus in the literature regarding the reliability between the DXA and ADP methods.

Thus, in lines 111-113, we inserted following wording: “In the present review, we consider the results of DXA and ADP as valid and reliable for determining body composition in the presented outcome (inserting 4 references)”.

Note: in the Discussion item, in the reliability studies, it was verified that the meta-analysis was able to mitigate the possible lack of agreement between the DXA and ADP instruments. We have inserted the following wording (lines 385-388): “Even admitting the possibility of inaccuracy between the DXA and PDA instruments, admitted as the reference standard, it seems that the meta-analysis was able to mitigate the incompatibility between them”.

2. Lines 44-47, read whether the observer used BodyMetrix™ to compare Three-Site and Seven-Site measurements, suggesting further refinement of the Seven-Site statistical analysis data.

Lines 44-47: “However, the P-US measures satisfactorily when it measures up to three sites associated with anthropometric perimeters in Multiple Linear Regression Equations with samples greater than 100 participants, and males”.

Reply to Reviewer 1:

The reports in lines 44-47 reinforce that the best results presented by the meta-analysis are those that used prediction equations involving the P-US measuring up to three sites associated with anthropometric measurements with samples greater than 100 male participants.

It is possible that we didn’t word it properly, so we apologize. Thus, in consideration of the Reviewer, we decided to change the wording: “The P-US associated with anthropometric perimeters satisfactorily measures the BF% with samples greater than 100 participants, and males”.

3. Line 154, The QUADAS-2 in the design method is proposed to be modified to QUADAS-C (Quality Assessment of Diagnostic Accuracy Studies-Comparative). The QUADAS-C tool was developed as an extension of QUADAS-2 to assess the risk of bias in comparative diagnostic test accuracy studies.

Line 154-158: QUADAS-2(24) was used to assess the risk of bias in the validity studies, which is structured so that the key domains (Patient Selection; Index Test(s); Reference Standard; Flow and Time) are classified in terms of risk of bias and research applicability concerns. Each key domain presents a series of guiding questions that help in judging biases and applicability, as shown in Figure 2.

Reply to Reviewer 1:

To respond to the Reviewer's request, we turned to Yang et al. (2020) and some published reviews (YOON et al., 2023; CHOW et al., 2023; TSIKOPOULOS et al., 2022; ZYFODIA et al., 2021) to support our position.

The QUADAS-C tool (Quality Assessment of Diagnostic Accuracy Studies-Comparative) was developed as an extension of QUADAS-2 to assess the risk of bias in comparative studies of diagnostic accuracy when evaluating the accuracy of 2 or more Index Tests compared to each other and to the Reference Standard. The “C” stands for comparison when assessing the risk of bias in studies included in a systematic review. Table 1, below, presents the characteristics of the aforementioned studies compared with our manuscript.

Table 1: Comparison between different publications

Author (year) Index Tests Reference Standard

 (ITa vs ITb) 

Yoon et al. (2023) ITa: AI Tomography

 ITb: Evaluation by Radiologists

Chow et al. (2023) ITa: PSMA-PET Histopathology

 ITb: CIM

Tsikopoulos et al. (2022) ITa: DTT Musculoskeletal Infection Society Criteria

 ITb: Sonification

Zyfodia et al. (2021) ITa: Xpert Ultra Rifacipin resistance

 ITb: Xpert MTB

Our manuscript ITa: P-US DXA or ADP or HW

 ITb:

ITa: Index Test a; ITb: Index Test b; AI: Artificial intelligence; PSMA-PET: positron emission tomography of prostate-specific membrane antigen; CIM: Conventional Imaging Modalities; DTT: dithiothreitol; Xpert Ultra: rapid test to detect tuberculosis; Xpert MTB: rapid test to detect tuberculosis; P-US: Portable Ultrasound; DXA: Dual-energy X-ray Absorptiometry; ADP: Air Displacement Plethysmography; HW: Hydrostatic Weighing.

Our manuscript uses only one IT (P-US), there are no comparisons with other index tests, we evaluate its results with the Reference Standard (DXA or ADP, or HW). By associating anthropometric perimeters with the P-US results in prediction equations, we do not consider this procedure as another instrument for evaluating BF%. We understand that QUADAS-C would be necessary if studies were located that evaluated the performance between P-US (ITa) and bioimpedance (ITb), having DXA as the Reference Standard, a hypothetical study that has not been located so far.

These being the arguments and with the necessary respect, we disagree with Reviewer 1's suggestion.

Minor Issues

4. Lines 100-101, 123-125, 218, refer to the website address as deleted or abbreviated as possible

Reply to Reviewer 1:

Because of the suggestions, we decided to remove the electronic addresses.

100-101 www.prisma-statment.org

123 www.decs.bvsalud.org/en/

125 https://www.ncbi.nlm.nih.gov/mesh/

5. In the flow diagram in Figure 1, the changes are corrected in the order of Identification、Screening、Eligibility、Included.

Reply to Reviewer 1:

Because of the suggestions, we have corrected the flowchart.

6. In Figure 3 and Figure 5, it is recommended to modify the setting position of Result in the figure, and do not overlap with other contents

Reply to Reviewer 1:

Taking into account the suggestions, we have improved the image.

REFERENCES

Yang B, Mallett S, Takwoingi Y, Davenport CF, Hyde CJ, Whiting PF, Deeks JJ, Leeflang MMG; QUADAS-C Group†; Bossuyt PMM, Brazzelli MG, Dinnes J, Gurusamy KS, Jones HE, Lange S, Langendam MW, Macaskill P, McInnes MDF, Reitsma JB, Rutjes AWS, Sinclair A, de Vet HCW, Virgili G, Wade R, Westwood ME. QUADAS-C: A Tool for Assessing Risk of Bias in Comparative Diagnostic Accuracy Studies. Ann Intern Med. 2021 Nov;174(11):1592-1599. doi: 10.7326/M21-2234. Epub 2021 Oct 26. PMID: 34698503.

Yoon JH, Strand F, Baltzer PAT, Conant EF, Gilbert FJ, Lehman CD, Morris EA, Mullen LA, Nishikawa RM, Sharma N, Vejborg I, Moy L, Mann RM. Standalone AI for Breast Cancer Detection at Screening Digital Mammography and Digital Breast Tomosynthesis: A Systematic Review and Meta-Analysis. Radiology. 2023 Jun;307(5):e222639. doi: 10.1148/radiol.222639. Epub 2023 May 23. PMID: 37219445; PMCID: PMC10315526.

Chow KM, So WZ, Lee HJ, Lee A, Yap DWT, Takwoingi Y, Tay KJ, Tuan J, Thang SP, Lam W, Yuen J, Lawrentschuk N, Hofman MS, Murphy DG, Chen K. Head-to-head Comparison of the Diagnostic Accuracy of Prostate-specific Membrane Antigen Positron Emission Tomography and Conventional Imaging Modalities for Initial Staging of Intermediate- to High-risk Prostate Cancer: A Systematic Review and Meta-analysis. Eur Urol. 2023 Jul;84(1):36-48. doi: 10.1016/j.eururo.2023.03.001. Epub 2023 Apr 7. PMID: 37032189.

Tsikopoulos K, Christofilos SI, Kitridis D, Sidiropoulos K, Stoikos PN, Gravalidis C, Givissis P, Papaioannidou P. Is sonication superior to dithiothreitol in diagnosis of periprosthetic joint infections? A meta-analysis. Int Orthop. 2022 Jun;46(6):1215-1224. doi: 10.1007/s00264-022-05350-z. Epub 2022 Feb 24. Erratum in: Int Orthop. 2022 Mar 24;: PMID: 35199219.

Zifodya JS, Kreniske JS, Schiller I, Kohli M, Dendukuri N, Schumacher SG, Ochodo EA, Haraka F, Zwerling AA, Pai M, Steingart KR, Horne DJ. Xpert Ultra versus Xpert MTB/RIF for pulmonary tuberculosis and rifampicin resistance in adults with presumptive pulmonary tuberculosis. Cochrane Database Syst Rev. 2021 Feb 22;2:CD009593. doi: 10.1002/14651858.CD009593.pub5. PMID: 33616229.

---

## [Decision Letter · Decision Letter 1]

12 Sep 2023

PONE-D-23-06958R1Validity and reliability of portable A-mode ultrasound in measuring body fat percentage: A systematic review with meta-analysisPLOS ONE

Dear Dr. Bomfim,

Thank you for submitting your manuscript to PLOS ONE. After careful consideration, we feel that it has merit but does not fully meet PLOS ONE’s publication criteria as it currently stands. Therefore, we invite you to submit a revised version of the manuscript that addresses the points raised during the review process.

ACADEMIC EDITOR:Dear Authors,your R1 manuscript version has been revised by one expert in the field that still retrieved some minor point you should consider during the revision process. Please submit your revised manuscript by Oct 27 2023 11:59PM. If you will need more time than this to complete your revisions, please reply to this message or contact the journal office at plosone@plos.org. Please include the following items when submitting your revised manuscript:A rebuttal letter that responds to each point raised by the academic editor and reviewer(s). You should upload this letter as a separate file labeled 'Response to Reviewers'.A marked-up copy of your manuscript that highlights changes made to the original version. You should upload this as a separate file labeled 'Revised Manuscript with Track Changes'.An unmarked version of your revised paper without tracked changes. You should upload this as a separate file labeled 'Manuscript'.If applicable, we recommend that you deposit your laboratory protocols in protocols.io to enhance the reproducibility of your results. Protocols.io assigns your protocol its own identifier (DOI) so that it can be cited independently in the future. For instructions see: https://journals.plos.org/plosone/s/submission-guidelines#loc-laboratory-protocols. Additionally, PLOS ONE offers an option for publishing peer-reviewed Lab Protocol articles, which describe protocols hosted on protocols.io. Read more information on sharing protocols at https://plos.org/protocols?utm_medium=editorial-email&utm_source=authorletters&utm_campaign=protocols.

We look forward to receiving your revised manuscript.

Kind regards,

Emiliano Cè

Academic Editor

PLOS ONE

Journal Requirements:

Reviewers' comments:

Reviewer's Responses to Questions

**Comments to the Author**

1. If the authors have adequately addressed your comments raised in a previous round of review and you feel that this manuscript is now acceptable for publication, you may indicate that here to bypass the “Comments to the Author” section, enter your conflict of interest statement in the “Confidential to Editor” section, and submit your "Accept" recommendation.

Reviewer #1: All comments have been addressed

2. Is the manuscript technically sound, and do the data support the conclusions?

Reviewer #1: Yes

3. Has the statistical analysis been performed appropriately and rigorously? 

Reviewer #1: I Don't Know

4. Have the authors made all data underlying the findings in their manuscript fully available?

Reviewer #1: Yes

5. Is the manuscript presented in an intelligible fashion and written in standard English?

Reviewer #1: Yes

6. Review Comments to the Author

Reviewer #1: The authors made some progress in the meta-analysis and systematic review of Bodymetrix® branded portable ultrasound (P-US) measurement of body fat percentage, supplementing our previous understanding of how to use P-US to measure body fat percentage and suggesting that we have important implications for better, faster and more accurate measurement of body fat percentage.

7. PLOS authors have the option to publish the peer review history of their article (what does this mean?). If published, this will include your full peer review and any attached files.

Reviewer #1: **Yes: **Hai Cui

---

## [Author Response · Author response to Decision Letter 1]

15 Sep 2023

In order to respond to the reviewer’s suggestions, the changes and arguments regarding the comments are presented below:

Comment from Reviewer #1:

1. If the authors have adequately addressed your comments raised in a previous round of review and you feel that this manuscript is now acceptable for publication, you may indicate that here to bypass the “Comments to the Author” section, enter your conflict of interest statement in the “Confidential to Editor” section, and submit your "Accept" recommendation.

Reviewer #1: All comments have been addressed.

Reply to Reviewer #1: thanks!

2. Is the manuscript technically sound, and do the data support the conclusions?

Reviewer #1: Yes

Reply to Reviewer #1: no comment

3. Has the statistical analysis been performed appropriately and rigorously?

Reviewer #1: I Don't Know

Reply to Reviewer #1: We inform you that the statistical analysis used was appropriate for the following reasons:

3.1. The JAMOVI program has already been used in other studies(1–5) and is capable of analyzing mean effect measures and correlations;

3.2. The selection of effect measures was appropriate: for validity, Pearson’s correlation coefficient; for reliability, the result of the Bland-Altman Test, which is the mean of the errors;

3.3. Heterogeneity was assessed for both outcomes, and according to its results, the analysis model (fixed effect or random effects) was used to incorporate it or not into the meta-analysis and explain it;

3.4. In the case of high and true heterogeneity, subgroup analyzes were performed in an attempt to explain it;

3.5. All of this information is clearly described in the Effect measures and Synthesis of collected data methods (lines 184-246) in the Method section.

4. Have the authors made all data underlying the findings in their manuscript fully available?

Reviewer #1: Yes

Reply to Reviewer #1: no comment.

5. Is the manuscript presented in an intelligible fashion and written in standard English?

Reviewer #1: Yes

Reply to Reviewer #1: no comment, thanks!

6. Review Comments to the Author 

Reviewer #1: The authors made some progress in the meta-analysis and systematic review of Bodymetrix® branded portable ultrasound (P-US) measurement of body fat percentage, supplementing our previous understanding of how to use P-US to measure body fat percentage and suggesting that we have important implications for better, faster and more accurate measurement of body fat percentage.

Reply to Reviewer #1: no comment.

7. PLOS authors have the option to publish the peer review history of their article (what does this mean?). If published, this will include your full peer review and any attached files.

Do you want your identity to be public for this peer review? For information about this choice, including consent withdrawal, please see our Privacy Policy.

Reviewer #1: Yes: Hai Cui

The authors made some progress in the meta-analysis and systematic review of Bodymetrix® branded portable ultrasound (P-US) measurement of body fat percentage, supplementing our previous understanding of how to use P-US to measure body fat percentage and suggesting that we have important implications for better, faster and more accurate measurement of body fat percentage. However, there are still some issues that need further improvement, as described below. 

Minor Issue

1. Lines 214-215, It is recommended to delete the connection to the JAMOVI 2.2.5 software program download site, as any update or change may lead to misunderstanding by readers or researchers.

Reply to Reviewer #1: in response to the suggestion, we decided to remove the JAMOVI software version and web address.

Please check "Response to Reviewer.docx"

2. Figure 2, Modify the color distinction in Figure 2 to be consistent with that in Table 1. Different risk levels are suggested to be expressed in numbers (e.g: low risk 1, high risk 2, unknown risk 3).

Reply to Reviewer #1: in response to the suggestion, we standardized the colors based on Table 1 and inserted the numbers 1, 2 and 3 for “low risk”, “high risk” and “uncertain risk”, respectively.

Please check "Response to Reviewer.docx"

3. Table 3, Change the data format of the Heterogeneity and Meta-analysis items in Table 3 to left-justified.

Reply to Reviewer #1: We changed the formatting of the table, as suggested.

Please check "Response to Reviewer.docx"

References:

1. Leporace G, Metsavaht L, Gonzalez FF, Arcanjo de Jesus F, Machado M, Celina Guadagnin E, et al. Validity and reliability of two-dimensional video-based assessment to measure joint angles during running: A systematic review and meta-analysis. J Biomech [Internet]. 2023;157(July):111747. Available at: https://doi.org/10.1016/j.jbiomech.2023.111747.

2. Dokponou YCH, Badirou OBA, Agada KN, Dossou MW, Lawson LD, Ossaga MAD, et al. Transcranial doppler in the non-invasive estimation of intracranial pressure in traumatic brain injury compared to other non-invasive methods in lower-middle income countries: Systematic review and meta-analysis. J Clin Neurosci. 2023;113(May):70–6. 

3. Pentapati KC, Yeturu SK, Siddiq H. A reliability generalization meta-analysis of Child Oral Impacts on Daily Performances (C – OIDP) questionnaire. J Oral Biol Craniofacial Res [Internet]. 2020;10(4):776–81. Available at: https://doi.org/10.1016/j.jobcr.2020.10.017

4. Farzan R, Hosseini SJ, Firooz M, Tabarian MS, Jamshidbeigi A, Samidoust P, et al. Perceived stigmatisation and reliability of questionnaire in the survivors with burns wound: A systematic review and meta-analysis. Int Wound J. 2023;(March):1–13. 

5. Olumade TJ, Uzairue LI. Clinical characteristics of 4499 COVID-19 patients in Africa: A meta-analysis. J Med Virol. 2021;93(5):3055–61.

---

## [Decision Letter · Decision Letter 2]

2 Oct 2023

Validity and reliability of portable A-mode ultrasound in measuring body fat percentage: A systematic review with meta-analysis

PONE-D-23-06958R2

Dear Dr. Bomfim,

We’re pleased to inform you that your manuscript has been judged scientifically suitable for publication and will be formally accepted for publication once it meets all outstanding technical requirements.

Kind regards,

Emiliano Cè

Academic Editor

PLOS ONE

Additional Editor Comments (optional):

Reviewers' comments:

Reviewer's Responses to Questions

**Comments to the Author**

1. If the authors have adequately addressed your comments raised in a previous round of review and you feel that this manuscript is now acceptable for publication, you may indicate that here to bypass the “Comments to the Author” section, enter your conflict of interest statement in the “Confidential to Editor” section, and submit your "Accept" recommendation.

Reviewer #1: All comments have been addressed

2. Is the manuscript technically sound, and do the data support the conclusions?

Reviewer #1: Yes

3. Has the statistical analysis been performed appropriately and rigorously? 

Reviewer #1: I Don't Know

4. Have the authors made all data underlying the findings in their manuscript fully available?

Reviewer #1: Yes

5. Is the manuscript presented in an intelligible fashion and written in standard English?

Reviewer #1: Yes

6. Review Comments to the Author

Reviewer #1: The authors made some progress in the meta-analysis and systematic review of Bodymetrix® branded portable ultrasound (P-US) measurement of body fat percentage, supplementing our previous understanding of how to use P-US to measure body fat percentage and suggesting that we have important implications for better, faster and more accurate measurement of body fat percentage.

7. PLOS authors have the option to publish the peer review history of their article (what does this mean?). If published, this will include your full peer review and any attached files.

Reviewer #1: **Yes: **HAI CUI

---

## [Editor Report · Acceptance letter]

5 Oct 2023

PONE-D-23-06958R2 

Validity and reliability of portable A-mode ultrasound in measuring body fat percentage: A systematic review with meta-analysis 

Dear Dr. Bomfim:

I'm pleased to inform you that your manuscript has been deemed suitable for publication in PLOS ONE. Congratulations! Your manuscript is now with our production department. 

Kind regards, 

on behalf of

Prof. Emiliano Cè 

Academic Editor

PLOS ONE